# BLIPEE: Faster BLIP with Adversarially Trained Early Exits

## Abstract

In recent years, Vision-Language Models (VLMs) have shown remarkable performance improvements in vision-language tasks. However, their large size poses challenges for real-world applications where inference latency is a concern. To tackle this issue, we propose employing Early Exit (EE) strategies in VLM. However, training exit classifiers in VLMs is challenging, particularly with limited labeled training data. To address this, we introduce BLIPEE, an adversarial training approach within a GAN-based framework. Here, each exit consists of a transformer layer and a classifier, and the transformer layer is adversarially trained to produce feature representations similar to the final layer, while a feature classifier serves as the discriminator. Our method focuses on performing input-adaptive inference that mitigates the overthinking issue and increases inference speed. Experimental results demonstrate the effectiveness of our approach in enhancing accuracy and model robustness by mitigating overthinking and the phenomenon of *mid-crisis* that we highlight. The anonymized source code is available at `https://anonymous.4open.science/status/BLIPEE-3ED3`.

## 1 Introduction

Vision-Language Pre-training (VLP) has evolved significantly with the emergence of sophisticated pre-trained Vision Language Models (VLMs). These models have consistently pushed the performance boundaries across various vision-language tasks. However, their demanding computational requirements and inference latency pose challenges for real-time applications. Several models, such as BLIP-2 Li et al. (2023), leverage off-the-shelf large-scale pre-trained models as building components with their parameters frozen. This reduces VLMs training parameters but makes them susceptible to issues related to overthinking during inference, as highlighted in previous studies Kaya et al. (2019); Zhou et al. (2020), leading to performance degradation and wasteful computation.

The use of the Language Model (LM) component with frozen parameters not only makes VLM susceptible to overthinking but also to another phenomenon that we term *mid-crisis*. This phenomenon occurs when intermediate layers suffer performance drops due to the search for irrelevant features. While initial layers capture shallow representations and syntactic information, and deep layers learn semantic-fusion relations Fei et al. (2022), intermediary layers tend to capture dataset patterns that degrade their performance, even losing the information learned by initial layers, and the model has to regain the lost information again in deeper layers. We illustrate this phenomenon in Figure 2 and Section 3.1, showcasing BLEU-4 Papineni et al. (2002) scores of exits in the COCO Lin et al. (2014) dataset used for image captioning. This raises the question: how can we mitigate mid-crisis and overthinking to enhance the accuracy and efficiency of VLMs?

We address this issue using 'Early Exit' (EE) techniques Xin et al. (2020); Zhou et al. (2020); Zhu (2021). It is a widely adopted input-adaptive technique that seeks to alleviate computational costs by bypassing certain layers while retaining the broad knowledge encoded in large-scale models. It is based on the fact that real-world datasets consist of 'easy' and 'hard' samples; hence, each does not require the same amount of computation. Implementing EE in VLMs introduces additional parameters requiring training, necessitating substantial amounts of labeled data, which can be prohibitively expensive. This limitation hampers the widespread adoption of EE methods in VLMs with good zero-shot capabilities.

We introduce a novel EE training technique named BLIPEE: Faster BLIP with Adversarially Trained EE designed to enhance the efficiency of VLMs while maintaining performance and speed with minimal training requirements. Our method leverages the capabilities of the VLMs to generate high-quality outputs alongside a feature classifier within a Generative Adversarial Network (GAN)-based framework Creswell et al. (2018) to minimize the discrepancy between feature representations at exit points and the final layer in an adversarial setup. This approach significantly reduces unnecessary computation at intermediate layers and elevates accuracy by learning feature representations directly from the final layer.

Our method is tailored to VLMs with exits and *Feature Classifiers* (FCs) attached to intermediary layers. Each exit consists of a single *exit transformer* (ET) and an *exit classifier* (EC). The exit transformer is a replica of the layers in the LM of the BLIP-2 model. They are used as generators and feature classifiers as discriminators in a GAN-based setup, as shown in Figure 1. The task of the feature classifier is to correctly classify if the input is from the exit or final layer, and the task of the exit transformer is to generate representations similar to the final layer to fool the feature classifier of that exit.

The exit classifiers at each of the exits are the same as that of the final layer classifier with frozen parameters and are used to map the outputs of exits to class probabilities. As the size of exit layer parameters is significantly smaller than that of exit classifiers, it substantially reduces the number of training parameters. In this way, a single LM layer attached to the exits helps produce similar feature representations and reduces the training parameters by utilizing the final layer classifier without training it at the exits. By attaching EEs, our method reduces the chances of overthinking and makes the inference process faster. Figure 3 shows how our method can speed up inference while maintaining comparable performance. The figure also provides a complete intuition of how our method can improve performance.

Adversarial training methods are susceptible to issues such as catastrophic forgetting Kirkpatrick et al. (2017); Ryu et al. (2022) and mode collapse. To circumvent these issues, we propose both supervised and unsupervised methods. In the supervised case, we use the hard labels from a small labeled dataset to remove the chances of catastrophic forgetting or exit training being stuck in a local optimum. In the unsupervised case, we replace the hard labels with the soft labels available as the output of vanilla BLIP-2, or we use the CapFilt Li et al. (2022b) used by the BLIP and BLIP-2 model to create high-quality synthetic labels.

- We introduce an EE strategy BLIPEE for VLMs to effectively mitigate inference latency by reducing unnecessary computations inherent in their large-scale architecture.
- To improve performance at EE classifiers, BLIPEE emulate the behaviour of the final layer at the exits through adversarial learning. To handle mode collapse and catastrophic forgetting, we propose supervised and unsupervised methods.
- Our model reduces the number of trainable parameters of the exits up to 37% by fixing the weights of the classifier attached to the exits, reducing the size of training dataset requirements.
- We experiment both qualitatively (see figure 3) and quantitatively on various tasks such as image captioning using COCO Lin et al. (2014) and NoCaps Agrawal et al. (2019) datasets and visual question-answering using VQAv2 Goyal et al. (2017), OK-VQA Marino et al. (2019), VizWiz Gurari et al. (2018) and GQA Hudson & Manning (2019) datasets. For visual dialogue, we use the VisDial Das et al. (2017) dataset. Our method provides inference speed $> 1.45\times$ with comparable and sometimes even better accuracy than vanilla BLIP-2 inference.

## 2 RELATED WORKS

We discuss the VLPs with LM components and EE strategies related to our work below.

**Vision-language Pre-training:** Different model architectures have been proposed for specific downstream tasks in VLPs, including dual-encoder architectures Radford et al. (2021); Jia et al. (2021), fusion encoder architectures Tan & Bansal (2019); Li et al. (2021), encoder-decoder architectures Cho et al. (2021); Wang et al. (2021); Chen et al. (2022b), and more recently, unified trans-

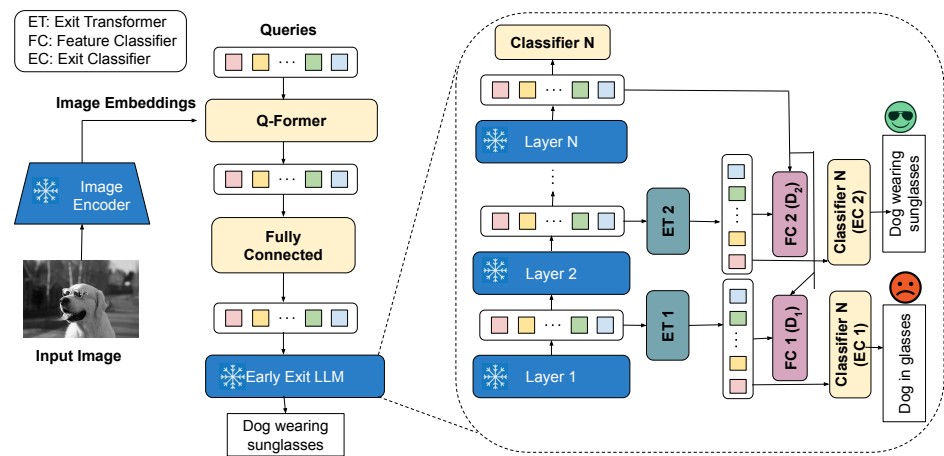

Figure 1: This figure shows the detailed architecture of our model. The Q-former output and previously predicted tokens are passed to the backbone. Then as the confidence threshold is met in the second exit, the sentence is predicted.

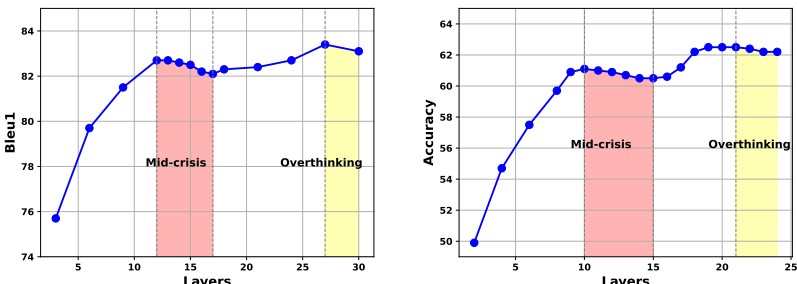

Figure 2: Left: BLEU-1 score for COCO with model BLIP-2-ViT-g-OPT$_{2.7B}$. Right: VQA accuracy on the VQAv2 dataset with BLIP-2-ViT-g-FlanT5$_{XL}$ showcasing mid-crisis and overthinking.

former architectures Li et al. (2022b); Wang et al. (2022b). Various pre-training objectives have also been introduced, such as image-text contrastive learning Radford et al. (2021); Yao et al. (2021); Li et al. (2021; 2022b), image-text matching Ju et al. (2021); Li et al. (2022b); Bao et al. (2021), and masked language modeling Li et al. (2021; 2022b); Yu et al. (2022); Wang et al. (2022b). However, these end-to-end models are inflexible to leverage readily available pre-trained models, such as LLMs Brown et al. (2020); Zhang et al. (2022a); Chung et al. (2024).

Recent approaches in vision-language pre-training have adopted the strategy of utilizing off-the-shelf pre-trained models and keeping them frozen during training. Some methods freeze only the image encoder Chen et al. (2020); Li et al. (2020); Zhang et al. (2021), and recent LiT Zhai et al. (2022) which uses a frozen pre-trained encoder for CLIP Radford et al. (2021)pre-training, while others freeze the language model to leverage knowledge from language-only pre-trained models for vision-to-language generation tasks Tsimpoukelli et al. (2021); Alayrac et al. (2022); Chen et al. (2022a); Mañas et al. (2022); Tiong et al. (2022); Guo et al. (2022). The primary challenge lies in aligning visual features with text space. To address this challenge, techniques like Frozen Tsimpoukelli et al. (2021) finetune image encoders or insert new cross-attention layers into language models to incorporate visual features. BLIP-2 Li et al. (2023) employs both frozen image encoders and language models for various vision-language tasks, achieving strong performance.

**Early Exits:** To minimize inference latency in deep neural networks, BranchyNet Teerapittayanon et al. (2016) introduced attaching exits classifiers at intermediary layers. This concept was extended by Shallow-deep Kaya et al. (2019), which effectively determines when to exit based on confidence

distribution at each exit classifier. Approaches like Huang et al. (2017); Yang et al. (2020); Han et al. (2023) improve EEs for image tasks by dynamically choosing different depths for different regions of the image. Additionally, approaches like Phuong & Lampert (2019) have employed knowledge distillation for image classification. For Natural Language Processing (NLP) tasks, several early exit frameworks have emerged Xin et al. (2020); Liu et al. (2021); Zhou et al. (2020); Liu et al. (2020); Wang et al. (2019; 2020); Zhu (2021); Ji et al. (2023); Balagansky & Gavrilov (2022); Zhang et al. (2022b); Bajpai & Hanawal (2024); Bajpai et al. (2023; 2024), primarily built on the BERT backbone. DeeCap Fei et al. (2022) introduces EE for image captioning tasks, employing an imitation network to replicate outputs from transformer layers within an encoder-decoder architecture. Similarly, MuE Tang et al. (2023) applies EE to OFA Wang et al. (2022a), a unified vision-language model tailored for multi-modal applications.

The key differences in our work are: 1) We employ adversarial training for efficient learning of EE models. 2) Our method can work under both semi-supervised and unsupervised setups by utilizing the zero-shot capabilities of the BLIP-2, while previous methods require a good amount of high-quality labeled training data, thus reducing size of training data.

## 3 METHODOLOGY

We start with a pre-trained BLIP-2 model that consists of three components: Image encoder, Q-former, and Language Model. We assume that LM consists of $N$ layers, and we attach exits to the $K$ chosen layers. Each exit consists of one Exit Transformer (ET) layer with the same configuration as one LM layer and an Exit Classifier (EC). The exit layer is such the parameters of the ET are trainable, and those of EC are frozen. We motivate our method by investigating the phenomenon of mid-crisis in BLIP-2.

### 3.1 MOTIVATION

In Fig. 2, we illustrate BLEU-1 score and VQA accuracy for various layers. As seen, performance dips at the middle layers after the initial improvement. This is due to the LM component, which is of large size and kept frozen during training. During pre-training, BLIP-2 aligns the features of text and images using the Q-former, the querying transformer. However, the Q-former provides the image-grounded text embeddings to the LM component in such a way that it produces high-quality predictions only at the final layer and not the intermediary layer. We use this observation to add a transformer layer at each exit instead of just adding a classifier at the exits. The transformer layer is used to mimic the behaviour of the final layer. In this way, we have low-level features already available at the initial layers and provide access to the high-level features at the exits by adversarially training them to produce representations similar to those of the final layer. This makes the exits more effective both in terms of accuracy and speeding up the inference.

We note that attaching a trainable classifier, as done in the previous works, can significantly increase the number of parameters as the size of the dictionary is large. For instance, a classifier attached at a single exit to BLIP-2 with OPT$_{2.7B}$ as a decoder has around $130M$ trainable parameters, and if we attach only 7 exits, this scales up to $900M$ parameters! In our methods, only the LM layer in each exit has trainable parameters. This adds $84M$ trainable parameters, scaling to $588M$ parameters for 7 exits. Hence, we reduce the training parameters by freezing the classifier weights and only training LM layer weights, thus reducing the trainable exit parameters by around $37\%$.

We next discuss our method which consists of two parts: 1) backbone finetuning and 2) exit training.

### 3.2 BACKBONE FINETUNING

The backbone is fine-tuned using the cross-entropy loss between the predicted token and the ground truth token. The loss function for fine-tuning could be formulated as:

$$\mathcal{L}(I) = \sum_{t=1}^{T} \log P_N(y_t^* | y_{1:t-1}^*, I) \tag{1}$$

where $I$ denotes the input image, $y_{1:T}^*$ is the true caption and $T$ is the caption length. $P_N$ denote the probability score from the final layer. In this step, the backbone learns to produce high-quality

features at the final layer and a classifier $C_N$ to map the feature representations at the final layers to class probabilities. Note that $C_N$ is part of the backbone. Once the fine-tuning is complete, we freeze the parameters of the backbone. This is done to maintain the optimal quality of the backbone.

## 3.3 Exits training

After fine-tuning the backbone, we attach $K$ exits to the LM component of the BLIP-2 model. We denote the set of exit indices by $[K]$. At each exit, we use a feature classifier $D_i$ that discriminates the feature representations of the transformer layer of the $i$th exit from that of the final layer. Specifically, it provides a score to an input feature representation, whether it is from the final layer. We have a separate feature classifier for every exit as feature representations at different exits can differ.

In our setup, the feature classifier acts as a discriminator, and the transformer layer exits as a generator; the goal of the transformer layer is to generate feature representations similar to that of the final layer. We train them alternately as in the original GAN framework. This training problem can be setup as an unconstrained optimization problem. Let $E_i$ denote the transformer layer in the $i$th exit. The feature classifier loss for an exit $i \in [K]$ and an input image $I$ could be formulated as:

$$\mathcal{L}_i^{fc}(h_t^i, h_t^N | y_{1:t-1}^*, I) = -\log D_i(h_t^N | y_{1:t-1}^*, I) - \log(1 - D_i(E_i(h_t^i | y_{1:t}^*, I))) \tag{2}$$

where $h_t^i$ is the feature (hidden) representation at $i$th layer and $h_t^N$ is the feature representation at the final layer of the LM for the $t$th token in the sentence. The overall loss across all exits could be written as $\mathcal{L}^{fc} = \sum_{i \in [K]} \mathcal{L}_i^{fc}$. It simultaneously updates the feature classifiers across all the exits.

The generator loss for the transformer layer in $i$th exit could be formulated as:

$$\mathcal{L}_i^{gen}(h_t^i | y_{1:t-1}^*, I) = -\log D_i(E_i(h_t^i | y_{1:t-1}^*, I))) \tag{3}$$

However, because the weights of the transformer layer of the exits are untied from the original backbone, they can face the issue of catastrophic forgetting or mode collapse. To circumvent these issues, we utilize the small labeled dataset to fine-tune the backbone. It guides the model to correct the learning trajectory and not let it get stuck to the local minima. The labeled data not only removes the issue of catastrophic forgetting but also helps in reducing overthinking as exits mimic the final layer and learn from the hard labels. The loss could be written as:

$$\mathcal{L}_i^{CE}(I, y_{1:t-1}^*) = \log P_i(y_t^* | y_{1:t-1}^*, I) \tag{4}$$

where $P_i$ denotes the probability score at the $i$th exit. The loss at exit $i$ will be $\mathcal{L}_i = \mathcal{L}_i^{CE} + \mathcal{L}_i^{gen}$. The overall loss for exits training will be $\mathcal{L} = \sum_{i \in [K]} \mathcal{L}_i$. After this step, the backbone is learned with attached exits.

## 3.4 Unsupervised setup

Recall from the previous section that labeled data was utilized to reduce the issue of catastrophic forgetting and mode collapse. As the BLIP-2 model has good zero-shot performance, we can utilize it to either distill the knowledge at the final layer or create a small set of pseudo labels to fulfill the requirements of the labeled dataset. We provide two methods for unsupervised learning.

**Using Knowledge Distillation:** In this case, we can directly utilize the soft labels from the final layer to distill the knowledge to the exits. The knowledge distillation loss for the $i$th layer could be formulated as:

$$\mathcal{L}_i^{KL} = KL(p_t^i, p_t^N) \tag{5}$$

where $p_t^i = C_N(E_i(h_t^i | y_{1:t-1}^*, I))$, $p_t^N = C_N(h_t^N | y_{1:t-1}^*, I)$ and $KL$ is the KL-divergence loss defined as $KL(p_t^i, p_t^N) = \sum_{v \in \mathcal{V}} p_t^i \log \frac{p_t^i}{p_t^N}$ where $\mathcal{V}$ is the vocabulary. We can train the exits by replacing the $\mathcal{L}_i^{CE}$ by $\mathcal{L}_i^{KL}$. Then the overall loss for exit $i$ is $\mathcal{L}_i = \mathcal{L}_i^{KL} + \mathcal{L}_i^{gen}$.

This method also utilizes the zero-shot capabilities of the BLIP-2 model. However, this method has a slightly lower performance than the CapFilt method proposed next, still it comes with lower computational cost and has comparable performance to vanilla BLIP-2 inference.

**Using CapFilt:** CapFilt Li et al. (2022b; 2023) is a method that is used in both the original BLIP and BLIP-2 models to generate high-quality synthetic captions. We use similar ideas to generate the

| Models | #Train Params | NoCaps Zero-shot | | | | | | | | Spd |
| | | in-domain | | near-domain | | out-domain | | full-dataset | | |
| | | C | S | C | S | C | S | C | S | |
|---|---|---|---|---|---|---|---|---|---|---|
| Vin VL | 345M | 102.9 | 14 | 94.8 | 13.7 | 88.1 | 11.9 | 95.1 | 13.2 | - |
| BLIP | 446M | 114.9 | 15.2 | 110.6 | 14.6 | 114.8 | 14.3 | 112.8 | 14.7 | - |
| SimVLM | 1.4B | 113.7 | 14.9 | 110.6 | 14.2 | 114.6 | 14.4 | 112.1 | 14.3 | - |
| BLIP-2 ViT $O_{2.7B}$ | 1.1B | 123.0 | 15.8 | 117.8 | 15.4 | 123.2 | 15.0 | 119.6 | 15.4 | 1.07× |
| BLIP-2 ViT $FT5_{XL}$ | 1.1B | 123.7 | 16.3 | **120.2** | **15.9** | 124.8 | 15.1 | 121.6 | 15.8 | 1.00× |
| *Early Exit models* | | | | | | | | | | |
| DeeBLIP | 1.8B | 115.2 | 15.3 | 111.5 | 14.7 | 115.4 | 14.5 | 112.4 | 14.5 | 1.41× |
| PABEE-BLIP | 1.8B | 117.7 | 15.4 | 114.2 | 14.8 | 117.6 | 14.7 | 112.9 | 14.6 | 1.29× |
| LeeBLIP | 1.8B | 119.4 | 15.5 | 115.8 | 14.8 | 120.1 | 14.9 | 116.3 | 15.1 | 1.38× |
| MuE | 1.8B | 118.1 | 15.4 | 115.3 | 14.8 | 118.7 | 14.8 | 114.8 | 14.9 | 1.44× |
| BLIPEE ViT $O_{2.7B}$ | 1.5B | 122.7 | 15.7 | 118.1 | 15.5 | 123.9 | 15.1 | 119.9 | 15.6 | **1.63×** |
| BLIPEE ViT $FT5_{XL}$ | 1.4B | **124.3** | **16.5** | 120.0 | **15.9** | **125.5** | **15.4** | **122.7** | **16.1** | 1.51× |

Table 1: Results on the Nocaps dataset during zero-shot transfer when the model is trained on the COCO dataset across various domains. Spd is the speedup achieved by the model. $O_{2.7B}$ is $OPT_{2.7B}$ and $FT5_{XL}$ is $FlanT5_{XL}$.

labeled dataset. In this step, a sample is passed through the BLIP-2 model, which then provides us with 10 possible captions for the given samples. We then use the CLIP ViT-L/14 model to rank the synthetic captions based on the image-text similarity produced by the CLIP model. We then keep the top-2 captions and keep them as synthetic captions that can be later utilized for training the exits by treating the synthetic captions as true captions. Creating synthetic captions using the CapFilt is more accurate but computationally heavy Li et al. (2022b).

## 3.5 Inference

We perform the caption in an autoregressive manner. This entails making a token-by-token prediction for a given image, where the layer at which the token is predicted is determined by the confidence score $S_i = \max_{v \in \mathcal{V}} C_N(E_i(h_t^i|y_{1:t-1}, I))(v)$ where $\mathcal{V}$ is the vocabulary. The input to the decoder is processed sequentially through the decoder layers until the confidence score $S_i$ is greater than a predefined threshold value $\alpha$. This threshold is set based on the accuracy-efficiency trade-off. The inference starts with the begin of the sentence token and the next token is predicted either at the exits or at the final layer. Then the predicted token is appended to the generated caption and passed as an input to the decoder for the next token prediction. The inference process stops when the end of the sentence token is predicted either at intermediary layers with a confidence score of more than $\alpha$ or at the final layer. Note that if the prediction confidence is below $\alpha$ for all the exits, then the sample is predicted at the final layer, irrespective of the confidence in the prediction.

## 4 Experiments

In this section, we provide details of all the experiments and some salient results of our work.

**Dataset and Metric:** We evaluate the performance of our method using the COCO Lin et al. (2014) and NoCaps dataset Agrawal et al. (2019) for image captioning. For Visual Question-answering tasks, we utilize the VQAv2 Goyal et al. (2017), OK-VQA Marino et al. (2019) and GQA Hudson & Manning (2019) datasets. For visual dialogue, we use the VisDial Das et al. (2017) dataset. We report key metrics including BLEU-4 Papineni et al. (2002), METEOR Banerjee & Lavie (2005), CIDEr Vedantam et al. (2015) and SPICE Anderson et al. (2016) scores for captioning. For VQA tasks, we report the VQA accuracy and for the Visual Dialog, we use the Mean Reciprocal Rank (MRR) Dai et al. (2024). To effectively consider the speedup, we define the speedup as inverse of the fraction of parameters used for inference on average. The speedup is formulated as:

$$\text{Speedup} = \frac{\text{Total parameters}}{\text{Average number of parameters used}}$$

| Models | #Train Params | VQAv2 train | VQAv2 test | Spd |
|---|---|---|---|---|
| *Without Exits* | | | | |
| ALBEF | 314M | 72.3 | 71.5 | - |
| BLIP | 385M | 73.9 | 72.1 | - |
| OFA | 930M | 75.7 | 75.6 | - |
| Flamingo80B | 10.6B | 77.9 | 78.1 | - |
| BLIP-2 V-O | 1.2B | 78.3 | 78.5 | $1.07\times$ |
| BLIP-2 V-F | 1.2B | 78.8 | 78.7 | $1.00\times$ |
| *Early Exit models (on BLIP-2)* | | | | |
| DeeBLIP | 1.9B | 75.4 | 75.9 | $1.52\times$ |
| PABEE-BLIP | 1.9B | 77.4 | 77.1 | $1.39\times$ |
| LeeBLIP | 1.9B | 78.1 | 77.8 | $1.65\times$ |
| BLIPEE-V-O | 1.6B | 78.7 | 79.0 | $\mathbf{1.77\times}$ |
| BLIPEE-V-F | 1.5B | **78.9** | **79.1** | $1.71\times$ |

Table 2: Results of semi-supervised application of our model to Visual-Question Answering tasks.

where the average number of parameters could be written as $\frac{1}{M}\sum_{I=1}^{M}\sum_{i=1}^{N_I} w_i \times (i+k) \times p$ where $p$ denotes the number of parameters in one layer of the LM component, $N_I$ denotes the number of predicted words in the caption for the image $I$, $M$ denotes the total number of input images and $k=1$ is the number of LM layers in the exit. Total parameters denote the total number of parameters in the backbone. The baseline for comparing speedup is BLIP-2 ViT-g FlanT5$_{\text{XL}}$. We only compare the speedup of early exiting methods and the BLIP-2 variants.

**Training:** In our setup, we utilize two variations of the BLIP-2 model with the same image encoder (ViT-g/14Dosovitskiy et al. (2020)) and frozen LLMs that are OPT-2.7B Zhang et al. (2022a) and FlanT5-XL Chung et al. (2024). We use the LAVIS Li et al. (2022a) library for implementation, training and evaluation. For training, we use the validation split of the datasets. We use 80% of validation split for training and the remaining 20% is reserved for development. We use labels of the validation dataset when the task is semi-supervised, else we just use the samples without labels. First, the backbone is fine-tuned for 10 epochs with a starting learning rate of 1e-5, which decays by 0.5 every 3 epochs. The backbone weights are then frozen post-fine-tuning and exits are attached to the backbone. We train exit weights for a further 3 epochs. We employ the Adam Kingma & Ba (2014) optimizer and a batch size of 16. For the feature classifier, we have used MLP with one hidden layer with a hidden state of size 3072 and a LeakyReLU Xu et al. (2015) activation function.

For the unsupervised tasks, we train the model for 3 epochs on the validation dataset (without labels) with knowledge distillation from the final layer. Optimizers and learning rates are kept the same as given above. Note that in CapFilt we apply the CapFilt method on the validation dataset (without labels) and generate synthetic labels. After this, we perform a similar procedure by treating the synthetic labels as the true labels as done for the semi-supervised tasks. Fine-tuning of the backbone is not required in an unsupervised setup.

**Inference:** Inference is conducted with a batch size of 1. We provide the results on the test dataset. For image captioning, we use a prompt as 'a photo of' as an initial input to the LM component. The threshold is chosen as the best-performing threshold from the set $\{0.5, 0.6, 0.7, 0.8, 0.9, 1.0\}$ on the held-out split of the validation dataset. More details on the hyperparameter setting can be found in Appendix 6 with the values of hyperparameters. All the experiments were performed with a combination of 6 GPU setups consisting of two NVIDIA RTX A6000 and four NVIDIA GeForce RTX 3080 Ti.

**Baselines:** We establish baseline models for performance evaluation, including vanilla BLIP-2 inference. Additionally, we compare with multimodal models VinVL Zhang et al. (2021), ALBEF Li et al. (2021), SimVLM Wang et al. (2021), OFA Wang et al. (2022a), and Flamingo Alayrac et al. (2022). We also assess state-of-the-art early exit methods originally proposed for the BERT backbone, such as DeeBERT Xin et al. (2020), PABEE Zhou et al. (2020), and LeeBERT Zhu (2021), adapted to the BLIP-2 backbone as DeeBLIP, PABEE-BLIP, and LeeBLIP, respectively. DeeBLIP uses confidence-based exiting, PABEE-BLIP employs patience-based exiting, and LeeBLIP combines knowledge distillation from the final layer with hard label learning. Furthermore, we apply

| Model | #Total params | VisDial test | VQAv2 train test | | OK-VQA test | GQA test | VizWiz test | Speed |
|---|---|---|---|---|---|---|---|---|
| *Without exits* | | | | | | | | |
| Flamingo3B | 3.2B | - | 53.2 | 49.4 | 41.5 | - | 28.9 | 1.28× |
| Flamingo9B | 9.3B | - | 55.7 | 51.8 | 44.7 | - | 28.8 | 0.44× |
| Flamingo80B | 80B | - | 59.1 | 56.2 | **50.4** | - | **31.5** | 0.05× |
| BLIP-2 ViT-g OPT$_{2.7B}$ | 3.8B | 34.1 | 54.6 | 52.0 | 31.2 | 34.2 | 27.0 | 1.07× |
| BLIP-2 ViT-g OPT$_{6.7B}$ | 7.8B | 37.5 | 55.9 | 53.7 | 36.1 | 36.4 | 27.2 | 0.52× |
| BLIP-2 ViT-g FlanT5$_{XL}$ | 4.1B | **45.9** | **64.9** | **62.5** | 40.6 | **44.5** | 29.8 | 1.00× |
| *Early Exit models (on BLIP-2 ViT-g FlanT5$_{XL}$)* | | | | | | | | |
| DeeBLIP | 4.7B | 33.4 | 41.3 | 42.8 | 23.4 | 27.8 | 20.1 | 1.39× |
| PABEE-BLIP | 4.7B | 35.7 | 49.6 | 51.3 | 31.2 | 34.3 | 23.6 | 1.31× |
| LeeBLIP | 4.7B | 39.1 | 57.7 | 57.1 | 37.1 | 39.7 | 26.4 | 1.29× |
| MuE | 4.7B | 36.6 | 55.4 | 53.6 | 33.7 | 37.1 | 24.7 | 1.36× |
| BLIPEE ViT-g OPT$_{2.7B}$ | 4.3B | 32.3 | 55.5 | 53.4 | 35.6 | 44.7 | 26.8 | **1.51×** |
| BLIPEE ViT-g FlanT5$_{XL}$ | 4.5B | **45.5** | **64.5** | **62.1** | **40.3** | **44.0** | **29.5** | 1.45× |

Table 3: Results of the unsupervised Visual-Question Answering and VisDial dataset. For VQA tasks, we report the VQA accuracy and for the visual dialogue, we report the Mean Reciprocal Rank(MRR).

| Models | COCO Karpathy test | | | | |
|---|---|---|---|---|---|
| | B@4 | C | S | M | Spd |
| OFA | 37.5 | 130.3 | 25.2 | 31.1 | - |
| Flamingo | 38.5 | 134.1 | 24.1 | 27.8 | - |
| SimVLM | 38.6 | 138.3 | 24.8 | 29.8 | - |
| BLIP-2-V-O | 41.7 | 139.8 | 25.5 | 30.5 | 1.07× |
| BLIP-2-V-F | 40.4 | 141.5 | 25.2 | 29.1 | 1.00× |
| *Early Exit models* | | | | | |
| DeeBLIP | 32.8 | 115.1 | 20.9 | 25.3 | 1.65× |
| PABEE-BLIP | 34.2 | 119.8 | 21.4 | 26.2 | 1.45× |
| LeeBLIP | 37.4 | 132.0 | 22.8 | 27.6 | 1.59× |
| MuE | 37.9 | 137.5 | 23.6 | 28.5 | 1.41× |
| BLIPEE-V-O | **41.9** | **142.5** | **25.2** | **30.8** | **1.75×** |

Table 4: Results of semi-supervised training on the Karpathy test split of the COCO dataset.

the MuE Tang et al. (2023) early exiting method to the BLIP-2 backbone, using exits from the better-performing BLIP-2 variant for our baselines.

## 5 RESULTS

**Visual Question Answering:** We provide results on unsupervised (see table 3) as well as semi-supervised setups (see table 2). We observe that our method outperforms all early exit methods in terms of accuracy and speedup even with less number of trainable parameters. We even outperform the vanilla BLIP-2 inference due to overthinking in the BLIP-2 backbone which is mitigated by our input-adaptive inference. We also provide results on an unsupervised visual dialogue dataset where the task is similar to VQA but there is an additional context before the question i.e. a dialogue history between the user and the model.

**Image Captioning:** We provide results of semi-supervised and unsupervised setup in table 1 and 4 respectively. We clearly outperform the existing models in terms of both accuracy as well as speedup. For the NoCaps dataset, the model is fine-tuned on the COCO dataset. The speedup for NoCaps dataset is lower as there is a domain change from the training which lowers the confidence in prediction taking more samples to deeper exits for inference.

We observe performance improvement over previous baselines as we attached exits to the BLIP-2 model and by performing input-adaptive inference, we perform better than the BLIP-2 model, and as BLIP-2 outperforms other models, BLIPEE also outperforms others. For the early exiting baselines on BLIP-2, we outperform them as we have an additional component in the exits rather than just a linear classifier which helps in better performance of exits in terms of both performance

and speedup. Note that there is a decrease in accuracy when we are in an unsupervised setup, as our model mimics the final layer hence some amount of overthinking still remains. Still, we are comparable to the BLIP-2 inference. On the other hand, the labeled dataset in semi-supervised tasks helps the model learn the hardness of the incoming sample. This helps the model to overcome the overthinking issue.

We have utilized two versions of the BLIP-2 model that have decoder as FlanT5$_{XL}$ and OPT$_{2.7B}$. We observe that the speedup in BLIP-2 with OPT$_{2.7B}$ was higher as there are more layers in this hence they are more susceptible to overthinking issues. The speedup for VQA tasks was higher as these tasks are simpler than image captioning tasks. We have not reported the speedup of the models other than the variants of BLIP-2 as they have different types of architectures. In terms of speedup, our objective is to make BLIP-2 faster.

In table 5, we have shown the result of using the CapFilt method to generate synthetic captions in absence of the labeled dataset. We have reported the CIDEr score over the NoCaps dataset. We can observe that the model has improved upon the performance using CapFilt and the speedup has significantly increased. This effect is due to the good quality captions that help the exits learn better, hence it outputs more samples early increasing the speedup as compared to knowledge distillation.

**Ablation study:** We perform a detailed ablation study analysing the behaviour of different components, the hyperparameter $\alpha$ and important qualitative analysis in the Appendix.

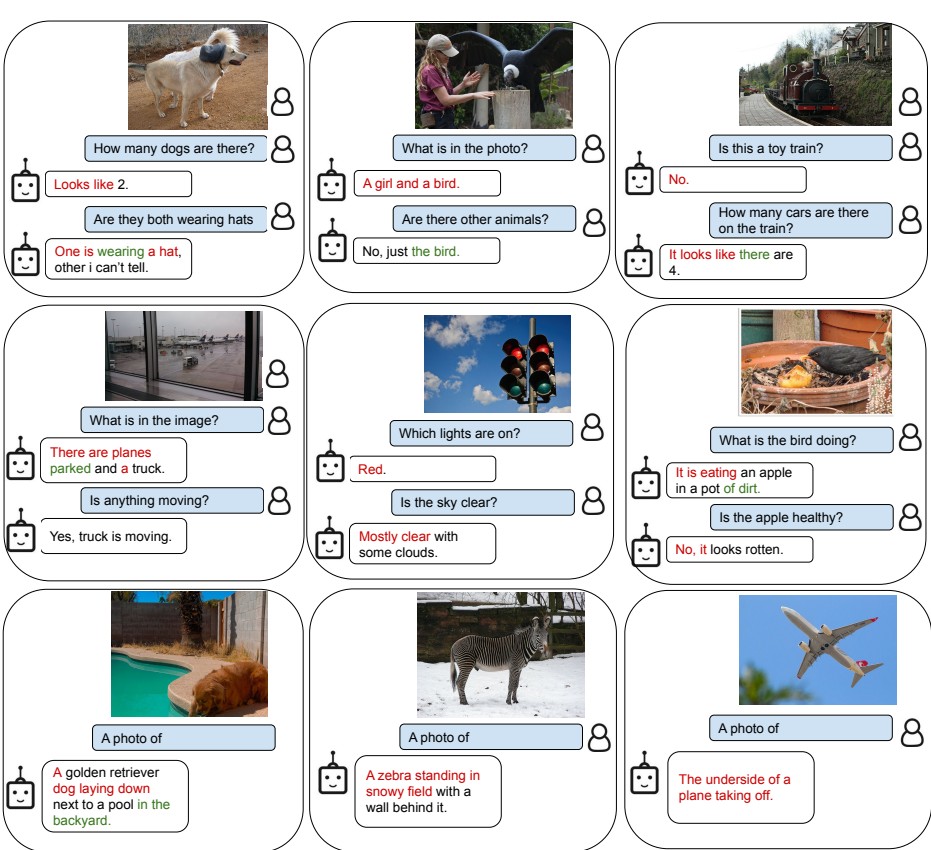

Figure 3: This figure provides some example outputs of BLIPEE ViT-g OPT$_{2.7B}$. The different colours show the difficulty levels of tokens in captions. Red: Easy to predict and predicted at initial (1-12) layers. Green: Mediocre hard, predicted at intermediary (13-24) layers. Black: Hard to predict, predicted at deeper (25-32) layers.

## 6 CONCLUSION

In this study, we introduced a novel inference technique BLIPEE, which leverages adversarial training of exits alongside the zero-shot capabilities of the BLIP-2 model. By employing BLIPEE, we effectively reduce the dependency on a vast amount of labeled training data typically required for exit training. Our approach involves adversarially training exits to generate representations similar to those of the final layer, thereby minimizing the need for extensive labeled data. Moreover, our exit design reduces the number of trainable parameters, resulting in lower computational costs. Experimental results demonstrate that our method significantly enhances inference speed while yielding high-quality outputs, occasionally even surpassing those produced by the final layer.

## 7 LIMITATIONS

For attaching exits to a large model such as BLIP-2, the crucial part is to decide where to attach exits within a given budget, i.e., what could be the best places for an exit in the LM component of the backbone without exceeding a certain amount of parameters. We answered that question by explaining the mid-crisis. However, the placements of exits with given budget criteria still remain unexplored which can make these models even faster within computational boundaries.

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

# A APPENDIX

## A.1 ABLATION STUDY

A detailed ablation study on the hyperparameter $\alpha$ modelling the accuracy and efficiency trade-off can be found in the Appendix A.2 and Fig 4b. In Fig 4a and section A.3 in the Appendix, we explain and verify the importance of different components in our model.

**Qualitative analysis:** In Fig 3, we provide some examples of the output provided by the BLIPEE model. The figure shows how the early exit models increase the speedup by predicting easier tokens earlier. For instance, the image in the last row and last column of the figure is an example of an easy sample where the tokens are predicted at initial layers. Similarly, for the image with a zebra, it can easily predict the easier token such as '*A zebra standing in a snowy field*' at the initial layers while the part of the image that is not easy to predict '*with a wall behind*' is predicted at deeper layers and predicting a high-quality caption overall while speeding up inference using the easiness of sample as well as token. Observe that there are fewer outputs from intermediate layers because of the mid-crisis phenomenon.

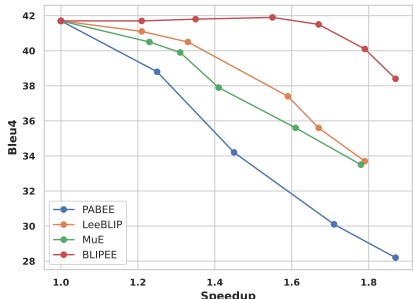

(a) Different combinations of loss function   (b) Speedup vs accuracy curve.

Figure 4: Left: BLEU-1 score for COCO with model BLIP-2-ViT-L-OPT$_{2.7B}$ with different components. Right: Speedup vs BLEU-4 curve for COCO dataset with ViT-L-OPT$_{2.7B}$.

| Model | No Caps Zero-shot | | | | |
|---|---|---|---|---|---|
| | in-domain | near-domain | out-domain | full-dataset | Spd |
| w/o CapFilt | 122.3 | 118.9 | 123.1 | 120.7 | 1.45× |
| Capfilt | 123.5 | 120.4 | 124.7 | 122.0 | 1.77× |

Table 5: Difference between CapFilt and Knowledge distillation method for an unsupervised setup.

## A.2   ACCURACY VS SPEEDUP

In figure 4b, we show the accuracy vs speedup curve which could be obtained by changing the threshold parameter $\alpha$. As we decrease the threshold parameter, samples exit from the initial layers even with less confidence, in this way all the samples are more prone to be incorrect decreasing the accuracy but as the threshold is decreased more samples exit from the initial layers and increase the speedup. One key observation is as we start decreasing the threshold, we observe that sometimes the performance even increases, this is the effect of overthinking, where some samples are correctly predicted at initial layers and might become wrong as they reach the final layer. We have also plotted the curves for other exiting methods and observed that our method has better stability as compared to other early exiting methods.

## A.3   IMPORTANCE OF DIFFERENT COMPONENTS

In figure 4a, we show the importance of different components of our method. We observe that there is a huge performance drop if we remove the knowledge distillation or cross-entropy loss from the overall loss function. This occurs due to catastrophic forgetting or mode collapse where the model gets stuck into local minima. On the other hand, if we remove the adversarial training part, there is again a performance drop, as we only train the classifier but we are not mapping the feature representations of the final layer and the exits hence exits only have low-level features which are insufficient to make correct predictions, hence resulting in a performance drop.

| LLM | OPT | FlanT5 |
|---|---|---|
| Exit Config | [3, 6, 9, 12, 24, 27, 30] | [3, 5, 7, 9, 12, 20, 22] |
| AdamW beta | [0.9, 0.999] | [0.9, 0.999] |
| Threshold | 0.8 | 0.8 |
| Inference beam size | 5 | 5 |
| Warmup Steps | 500 | 500 |

Table 6: More hyperparameter details of BLIPEE on different LM component in the BLIP-2 model. Note that the thresholds are chosen from the set $\{0.5, 0.6, 0.7, 0.8, 0.9, 1.0\}$

