# OpenReview forum: "BLIPEE:  Fast and Robust BLIP with Adversarially Trained Early Exits"
_ICLR.cc/2025/Conference — ICLR 2025 Conference Withdrawn Submission_

### Official Review · Reviewer_cURp · 2024-10-31

**Soundness:** 2
**Presentation:** 3
**Contribution:** 2
**Rating:** 5
**Confidence:** 4

**Summary:**

This paper proposes an early exit strategy to reduce the inference latency in Vision-Language Models. An adversarial training network within a GAN-based framework BLIPEE is utilized to reduce the negative impact of limited labeled training data. In the BLIPEE network, each exit contains a transformer layer and a classifier. The used input-adaptive inference mitigates the overthinking issue and increases inference speed. Experimental results show the effectiveness of the proposed BLIPEE. Authors provide anonymized source codes.

**Strengths:**

Some codes are provided to increase credibility of the BLIPEE network.

Various results show the effectiveness of the BLIPEE network. The designed method can improve the inference speed while yielding high-quality outputs.

Tables and figures are clear. I can understand them easily.

Limitation section is provided to present the work comprehensively.

**Weaknesses:**

The compared methods are not state-of-the-art. The newest compared methods (OFA and Flamingo) are published in 2022. Some state-of-the-art works are required for comparison.

In Table 2, BLIPEE-V-O and BLIPEE-V-F contain more Train Params than BLIP-2 V-O and BLIP-2 V-F. Why BLIPEE-V-O and BLIPEE-V-F have higher Spd than BLIP-2 V-O and BLIP-2 V-F?

In Figure 2, "Layers" should be "Layer number".

Some references needs to be revised, such as Li et al. (2020) in Line 634-642.

Some grammatical errors, such as "P_N denote the probability score ..." in Line 215.

**Questions:**

Please address the weakness.

---

> ### Author Response · Authors · 2024-11-22
> **Rebuttal**
>
> Thanks for the insightful comments.
>
> Ques-1: The compared methods are not state-of-the-art. The newest compared methods (OFA and Flamingo) are published in 2022. Some state-of-the-art works are required for comparison.
>
> Ans-1 The reason of not comparing with the SOTA methods is: 1) We are not proposing a new VLM that can beat SOTA. 2) Our objective is to fasten the existing BLIP-2 VLM model. We chose BLIP-2 as it provides flexibility to use any encoder and decoder. 3) We have compared with OFA and Flamingo as they are compared in BLIP-2 just to show that even after becoming faster our method has not reduced the performance of BLIP-2.
>
>
>
> Ques-2 : In Table 2, BLIPEE-V-O and BLIPEE-V-F contain more Train Params than BLIP-2 V-O and BLIP-2 V-F. Why BLIPEE-V-O and BLIPEE-V-F have higher Spd than BLIP-2 V-O and BLIP-2 V-F?
>
> Ans-2: We have added exits to the intermediate layers that adds on to the parameters of the backbone during training. But during inference, not all layers are required for prediction and a sample can exit earlier before even reaching till the final layer. As soon as a sample passes through a layer with an exit, the confidence of that layer in the prediction is checked and if the exit is confident enough, the sample exits the backbone without going deeper into the backbone. Thai speeds up the inference process.
>
>
> Presentations issues: In Figure 2, "Layers" should be "Layer number".
> Some references needs to be revised, such as Li et al. (2020) in Line 634-642.
> Some grammatical errors, such as "P_N denote the probability score ..." in Line 215
> Thanks for pointing out, we will fix these in the final version.
>
> We hope that we clarified most of your doubts, if you have any further questions, please let us know. If not please consider reassessing the scores as it seems you liked our work except some minor issues that can be easily fixed.

---

> > ### Comment · Reviewer_cURp · 2024-12-03
> > **I concern that this presented method can only handle a small number of cases. I lower my rating from 6 to 5.**
> >
> > About "Our objective is to fasten the existing BLIP-2 VLM model" in Ans-1, I think the objective is not enough for ICLR since the work is only used to fasten the specific model. At first, I thought this paper reconstructed BLIP-2 VLM model. Besides, I agree with Reviewer g7un: "the novelty for combining them in a unique way" is not enough for ICLR.
> >
> > About Ans-2, as shown in the tables, the speed increase is not amazing, which means that the early exit mechanism is not suitable for most cases.
> >
> > Thus, I concern that this presented method can only handle a small number of cases. I lower my rating from 6 to 5.

---

### Official Review · Reviewer_g7un · 2024-11-03

**Soundness:** 3
**Presentation:** 3
**Contribution:** 2
**Rating:** 3
**Confidence:** 4

**Summary:**

An EE strategy BLIPEE for VLMs to effectively mitigate inference latency by reducing unnecessary computations. BLIPEE emulate the behaviour of the final layer at the exits through adversarial learning. Experimental results demonstrate the effectiveness of this approach in enhancing accuracy and model robustness by mitigating overthinking and the phenomenon of mid-crisis that we highlight.

**Strengths:**

The key differences in this work are clear: 1) We employ adversarial training for efficient learning of EE models. 2) Our method can work under both semi-supervised and unsupervised setups by utilizing the zero-shot capabilities of the BLIP-2, while previous methods require a good amount of high-quality labeled training data, thus reducing size of training data.

**Weaknesses:**

1、Confusing symbols in fig1, why Classifier N, what’s meaning of D1/D2?
2、Missing some improtant references:
[1] NEO-KD: Knowledge-Distillation-Based Adversarial Training for Robust Multi-Exit Neural Networks
[2] L. Qendro and C. Mascolo, "Towards Adversarial Robustness with Early Exit Ensembles," 2022 44th Annual International Conference of the IEEE Engineering in Medicine & Biology Society (EMBC), Glasgow, Scotland, United Kingdom, 2022, pp. 313-316, doi: 10.1109/EMBC48229.2022.9871347.
3. Unsupervised manner is not novel, just self-labeling, but their pseudo labels are not accurate enough in general.
4. Ablation study is not enough, whether or not the proposed adversarial training is necessary?
5. Technical contributions are not enough, no matter of adversarial training or knowledge distillation are very commonly-used skills, no more new techniques are founded.

**Questions:**

1. The proposed early exit method was only tested on blip2, and other models were not tested, so the generalization abilty of the method cannot be confirmed. At the same time, the section 3.1 mentioned that the problem discussed in this paper is because Q-former generates image-grounded text embeddings. However, in the case that most VLMs do not use Q-former nowadays, is this method still applicable?
2. The paper tested VQA and caption tasks, and blip2 also tested the retrieval task. How does the method in this paper performs on the retrieval task?
3. The speedup in the article is calculated based on the number of parameters. Can it be calculated based on the actual inference time?

---

> ### Author Response · Authors · 2024-11-22
> **Rebuttal**
>
> Thanks for the insightful comments.
>
> Ques-1: Confusing symbols in fig1, why Classifier N, what’s meaning of D1/D2?
>
> Ans-1: Classifier N means the final classifier that is the classifier attached after the final layer of the model. D_1/D_2 are the discriminators at different layers at layers 1 /2  respectively.
>
>
> Ques-2 : Unsupervised manner is not novel, just self-labeling, but their pseudo labels are not accurate enough in general.
>
> Ans-2 Yes, the pseudo labels might not be very much accurate in general but they re a very good substitute of the true labels. Also, please note that we are not claiming novelty in the fact that we proposed the method of self-labeling but we are claiming novelty that we are the first ones to use it for an early exit model. Learning in unsupervised manner is a major problem in EEVLMs as most of the VLMs have good zero-shot capabilities so do not require training data. But as we add exits to initial layers there is a requirement of training data, it restricts the exit attachment to the backbone and making the process fast. To address this, we proposed BLIPEE that can even perform comparably when there is less or no training dataset.
>
> Ques-3: Ablation study is not enough, whether or not the proposed adversarial training is necessary?
>
> Ans-3: The necessity of adversarial training is justified by the objective that we need to meet. We want to generate samples from a given distribution with a given architecture, this is a similar setup to GAN where we need to generate images from a given distribution. If we remove the adversarial part, the method boils down to DeeBLIP, PABEE-BLIP methods already added in our baselines.
>
>
> Ques-5 :  Technical contributions are not enough, no matter of adversarial training or knowledge distillation are very commonly-used skills, no more new techniques are founded.
>
> Ans-5: Yes, we agree that these are the already existing approaches, but we claim the novelty for combining them in a unique way so that the overall method can solve the issues of latency in the VLMs without higher loss in accuracy.
>
> Ques-6: The proposed early exit method was only tested on blip2, and other models were not tested, so the generalization abilty of the method cannot be confirmed. At the same time, the section 3.1 mentioned that the problem discussed in this paper is because Q-former generates image-grounded text embeddings. However, in the case that most VLMs do not use Q-former nowadays, is this method still applicable?
>
> Ans-6: We chose BLIP-2 for experiments in our method as BLIP-2 model gives us the flexibility to use any encoder and decoder. As our method applies to the decoder in VLMs and we had to show results on multiple decoder, we have used BLIP-2. Yes, the problem might be specific to BLIP-2 as it uses the frozen decoder that might be more prone to mid-crisis.
> Note that our method can easily be extended to any VLM and could be appended to any of the VLM’s decoder as it does not have anything that is specific to the BLIP-2 model.
>
>
> Ques-7:  The paper tested VQA and caption tasks, and blip2 also tested the retrieval task. How does the method in this paper performs on the retrieval task?
>
> Ans-7: Note that our method only applies to the decoder of the BLIP-2 model. Image text retrieval task does not require the decoder which in turn reduces to vanilla BLIP-2 inference. Hence, we have not added those results.
>
>
> Ques-8: The speedup in the article is calculated based on the number of parameters. Can it be calculated based on the actual inference time?
>
> Ans-8: Yes, it can be calculated using the actual inference time but as all other existing work show speedup metric only as it can be easily converted to multiple metrics such as expected time reduction rate, average number of layers required for a dataset etc so to be fair and consistent we used the speedup to access the increase in speed during inference.
>
>
> We hope that we clarified most of your doubts. If you any any further questions, please let us know, else, we request you to consider reassessing the scores.

---

### Official Review · Reviewer_ofR1 · 2024-11-03

**Soundness:** 2
**Presentation:** 2
**Contribution:** 3
**Rating:** 5
**Confidence:** 3

**Summary:**

The paper presents an early exit strategy for training VLMs. The key idea is to attach exits across different language model layers, with each exit consisting of a transformer layer and a classifier. The transformer layers and the classifiers are trained through a GAN-based framework such that the transformer layers generate feature representations similar to the last layer. Training consists of (1) backbone fine-tuning and (2) exit training. For exit training, a semi-supervised setup and an unsupervised setup are discussed to train the transformer layer to generate features similar to the final layers. During inference, captions are generated in an autoregressive manner. Experimental results show that the proposed method outperforms prior early exit methods with less computational cost.

**Strengths:**

* The motivation is clear; the mid-crisis and overthinking phenomenon is intriguing.
* Early exit could also provide some insights into the reasoning mechanisms of LLMs, as shown in Figure 3.
* The idea is novel; using a GAN-based method for early exit is interesting and seems effective.

**Weaknesses:**

* The motivation for backbone fine-tuning is unclear and not explained. Why not use a pre-trained backbone? Does it help early exit?
* Most of the baselines come from earlier works. The baseline from recent VLM works, such as LLaVA, miniGPT-4, etc. are missing.
* According to Tables 1 and 2, the performance improvement seems incremental. Instead of the speedup calculated from L323, what is the speedup on the hardware specify in the paper? Does actual speedup align with this calculation?

Minor issue: Citations within the text are strange, check the formatting instruction.

**Questions:**

1. What is the speedup on hardware? Does early exit also speedup the causal self-attention (autoregressive) model?
2. What is $w_i$ in L342?
3. In Eqn (2), the last $y*$ should be $y*_{1:t-1}$?
4. Does other GAN-framework work? Such as WGAN?

---

> ### Author Response · Authors · 2024-11-22
> **Rebuttal**
>
> Thanks for the insightful comments.
>
> Que-1: The motivation for backbone fine-tuning is unclear and not explained. Why not use a pre-trained backbone? Does it help early exit?
>
> Ans-1: The backbone fine-tuning is required as the additional transformer layer in the exits needs to be trained for that a fine-tuning is required. If randomly initialized the performance came out to be nearly zero for initial layers. Hence a fine-tuning is required for most of the early exit methods.
>
>
>
> Que- 2: Most of the baselines come from earlier works. The baseline from recent VLM works, such as LLaVA, miniGPT-4, etc. are missing.
>
> Ans-2: As in our method, we are adding exits to the BLIP-2 decoder models, the main baselines are existing early exit methods and the vanilla BLIP-2 inference, we have added other baselines just to give a sense that BLIP-2 is faster with EE methods with comparable performance. We cannot directly compare the existing baselines with an EE model, but the major baseline is how fast the model inference is with what loss in performance as compared to existing EE methods and the vanilla inference. The main reason of this argument is that we are proposing a EE method and not a VLM backbone, hence we have not considered them as baselines. We have compared with OFA and Flamingo as they are compared in BLIP-2 just to show that even after becoming faster our method has not reduced the performance of BLIP-2
>
>
>
> Ques-3: According to Tables 1 and 2, the performance improvement seems incremental. Instead of the speedup calculated from L323, what is the speedup on the hardware specify in the paper? Does actual speedup align with this calculation?
>
> Ans-3: Note that our method is pushing two metrics simultaneously due to which the performance needs to be judged in both ways. Our goal is to reduce the trade-off impact of accuracy and efficiency i.e., the model provides faster inference while having the smallest decrement in the performance. In all the existing methods, we are the ones that best do it and here we claim the novelty and observe that in terms of speedup our method is better than existing methods and in terms of performance the drop is minimal.
>
> Speedup is a standard metric used for measuring the efficiency of EE models; it has already been used in various existing works. Speedup is proportional to any kind of hardware, which helps in a fair comparison as the actual hardware time might vary in various runs. This has been already explained in previous literature.
>
>
> Ques-4: What is the speedup on hardware? Does early exit also speedup the causal self-attention (autoregressive) model?
>
> Ans-4: The speedup metric reported is proportional to hardware however, we will report the actual time but the execution might take some time. Yes, in the OPT model which is autoregressive in generating text where we attached exits shows speedup in performance.
>
> $W_i$ denote the number of words that exit from the ith exit of the decoder.
>
>
> Ques-5: In Eqn (2), the last y∗ should be y∗1:t−1?
>
> Ans-5 Yes, that is a typo, thanks for pointing it out, we will fix this in the final version.
>
>
> Ques-6: Does other GAN-framework work? Such as WGAN?
>
> Ans-6 We have not explored that yet, but we intuitively sense that any GAN method can perform well that can generate high-quality features using its framework with a simple generator in our case which is a single transformer layer.
>
> We hope that we clarified most of your doubts. If you have any further questions, please let us know, else please consider reassessing the scores.

---

> > ### Comment · Reviewer_ofR1 · 2024-11-24
> > **Thanks for reply**
> >
> > Is it possible to conduct an experiment using the specified hardware mentioned in the paper to compare the theoretical speedup with the actual speedup? Thank you!

---

> > > ### Author Response · Authors · 2024-11-24
> > > **Reduction time**
> > >
> > > Thanks for the reply
> > >
> > > |            | True time | Expected time reduction | Speedup |
> > > | ---------- | --------- | ----------------------- | ------- |
> > > | Final exit | 660.2     | 1                       | 1       |
> > > | Ours       | 297.8     | \-42.8\%                  | 1.75x   |
> > >
> > > This table shows the true time expected time reduction and speedup, please note that they are close to each other. Here the dataset is the coco dataset.
> > >
> > > We are happy to take further questions, if not please consider reassessing the scores.

---

> ### Comment · Reviewer_ofR1 · 2024-11-24
>
> could you please provide more details about the experiment as well? Final exit is 660.2 and early exit is 297.8 which seems much better than theoretical speedup.

---

> > ### Author Response · Authors · 2024-11-25
> > **Further details**
> >
> > Sure
> >
> > The model used is BLIP-2ViT- FlanT5 xl and other setups are very same as detailed in the paper, we performed inference on the MScoco dataset with attached exits and got these results. Actually the speedup and time reduction both are consistent i.e., they both do not deviate much. If you need any specific info, please let us know.
> >
> > Thanks

---

### Official Review · Reviewer_KQvs · 2024-11-04

**Soundness:** 3
**Presentation:** 3
**Contribution:** 3
**Rating:** 5
**Confidence:** 4

**Summary:**

This paper proposes a very interesting issue of early exists on existing VLM models. To achieve this goal, it introduces an adversarial training approach within a GAN-based framework. Specifically, a transformer layer is utilized to mimic the output features of original VLM, and a classifier is utilized to determine when to exist. Experiments demonstrate the effectiveness of the proposed method.

**Strengths:**

1. The motivation of this paper is valuable.

2. This paper is easy to read and well-written.

3. The proposed components are reasonable.

**Weaknesses:**

1. The topic is too limited. The early exits issue is a good question for the existing VLM models, especially for large VLM models. The authors just implement the early exits strategy on a single BLIP model, limiting its scalability.

2. The illustrations of the proposed components are not clear. The authors should re-organize the method section for better presentation. A algorithmic pseudo-code of the entire process should also be provided.

3. The experiments are insufficient. The authors compare the efficiency of their BLIPEE with other large VLM like Flamingo, however, the authors do not apply their EE strategy to Flamingo for “plug-and-play” comparison.

4. Since the proposed method relies on additional transformer and classify layers, the authors should provide the comparison on model complexity.

**Questions:**

1. The topic is too limited. The early exits issue is a good question for the existing VLM models, especially for large VLM models. The authors just implement the early exits strategy on a single BLIP model, limiting its scalability.

2. The illustrations of the proposed components are not clear. The authors should re-organize the method section for better presentation. A algorithmic pseudo-code of the entire process should also be provided.

3. The experiments are insufficient. The authors compare the efficiency of their BLIPEE with other large VLM like Flamingo, however, the authors do not apply their EE strategy to Flamingo for “plug-and-play” comparison.

4. Since the proposed method relies on additional transformer and classify layers, the authors should provide the comparison on model complexity.

---

> ### Author Response · Authors · 2024-11-22
> **Rebuttal**
>
> Thanks for the insightful comments.
>
> Que 1: The topic is too limited. The early exits issue is a good question for the existing VLM models, especially for large VLM models. The authors just implement the early exits strategy on a single BLIP model, limiting its scalability.
>
> Ans 1: Yes, our method is scalable to multiple existing VLMs, however, we chose to perform experiments on BLIP-2 model as BLIP-2 provides the flexibility to use a wide variety of encoder and decoder models, such as we have used OPT 2.7B and 6.7 B and FlanT5 models as the decoder in our method. This makes our method general to any kind of VLM as they also consider a combination of encoder and decoder models and the idea is to show that our method can work under such scenarios, hence, we have used BLIP-2 that provides an option to use multiple decoders.
>
>
> Que-2: The illustrations of the proposed components are not clear. The authors should re-organize the method section for better presentation. A algorithmic pseudo-code of the entire process should also be provided.
>
> Ans-2: Sure, we will take up your suggestion and add an algorithm pseudocode into the final version of the paper, we will further clarify the use of the different components of our paper.
>
>
> Que - 3: The experiments are insufficient. The authors compare the efficiency of their BLIPEE with other large VLM like Flamingo, however, the authors do not apply their EE strategy to Flamingo for “plug-and-play” comparison.
>
> Ans - 3: Existing works such as DeeBERT, PABEE only compare against the early exiting baselines and the original model (comparing against the vanilla BERT and  existing EE approaches applied to BERT and not against GPT-2, LLama kind of models) that motivates us to only compare our approach against BLIP-2 vanilla inference and other existing EE methods applied to BLIP-2 . We have compared with OFA and Frozen as they are compared in BLIP-2 just to show that even after becoming faster our method has not reduced the performance of
> BLIP-2.
>
> Please note that we propose an EE method and not a VLM itself hence comparison is against the existing EE methods.
>
>
> Que-4: Since the proposed method relies on additional transformer and classify layers, the authors should provide the comparison on model complexity.
>
> Ans-4: We have provided the comparison on computational complexity, the speedup reported is a metric that considers the computational complexity of our method. The speedup metric is proportional to the computational requirements of the model. We have also added the parameter cost of additional exits during the speedup calculation. Not only this, we have also reported the model size.
>
> We hope that we clarified most of your doubts. If you have any further questions please let us know, else please consider reassessing the scores.

---

### Author Response · Authors · 2024-11-24
**Reminder**

Dear reviewers,

It is a gentle reminder to acknowledge our rebuttal and make changes accordingly.

Regards,

Authors

---

> ### Author Response · Authors · 2024-11-26
> **Reminder**
>
> Dear reviewers,
>
> It is a gentle reminder to acknowledge our rebuttal and make changes accordingly.
>
> Regards,
>
> Authors

---

### Author Response · Authors · 2024-11-30
**Reminder**

Dear reviewers,

It is a gentle reminder to acknowledge our rebuttal and make changes accordingly.

Regards,

Authors

---

### Note · Authors · 2024-12-16

I have read and agree with the venue's withdrawal policy on behalf of myself and my co-authors.